



# Effects of environmental and management factors on worldwide maize and soybean yields over the 20[th] and 21[st] centuries

Tzu-Shun Lin[1], Yang Song[2], Atul K. Jain[1], Peter Lawrence[3], Haroon S. Kheshgi[4]

[1]Department of Atmospheric Sciences, University of Illinois, Urbana, IL, 61801, USA
[2]Department of Hydrology and Atmospheric Sciences, University of Arizona, Tucson, AZ, 85721, USA
[3]National Center for Atmospheric Research, Boulder, CO, 80305, USA
[4]ExxonMobil Research and Engineering Company, Annandale, NJ, 08801, USA

*Correspondence to*: Atul K. Jain (jain1@illinois.edu)

**Abstract.** The land process model, ISAM, is extended to accurately simulate contemporary soybean and maize crop yields,
and estimate changes in yield over the period 1901-2100 driven by past and future changes in environmental factors --
atmospheric CO2 level ([CO2]) and climate (temperature and precipitation) – and management factors – nitrogen fertilizer and
deposition, irrigation, and crop harvest areas. Over the 20th century, each of these factors contributes to the increase in global
crop yield with increasing nitrogen fertilizer application the strongest of these drivers for maize and increasing [CO2] the
strongest for soybean. Over the 21st century, two future scenarios – RCP4.5-SSP2 and RCP8.5-SSP5 – of the environmental
and management factors are modeled to estimate their influence on future crop yield. For both crops under both scenarios,
changing climate drives yield lower, while rising [CO2] drives yield higher. For soybean, the negative climate effect is more
than offset by the other drivers -- particularly the increase in [CO2] – leading to an increase in global soybean yield by the
2090s.  For maize, combined negative climate and harvest area effects are offset in RCP4.5-SSP2, which has continued growth
in nitrogen fertilizer application, leaving global yield roughly unchanged. However, in RCP8.5-SSP5 maize yield declines
since this scenario has greater warming of climate and weaker nitrogen fertilizer application than RCP4.5-SSP2. The model
also projects differences between geographical regions; notably, higher temperatures in tropical regions limit photosynthesis
rates and reduce light interception by accelerating phenological development in both crops, particularly for RCP8.5-SSP5 and
for soybean.

## 1 Introduction

Crop yield has and will be affected by the environmental factors, such as atmospheric carbon dioxide level ([CO$_2$]), and
changes in temperature and precipitation patterns. However, the magnitude of these effects remains uncertain and
understanding of the interactions of environmental factors with crop management practices (e.g. irrigation and nitrogen (N)
fertilizer inputs) remains incomplete. The Global Gridded Crop Model Intercomparison (GGCMI) of Agricultural Model
Intercomparison and Improvement Project (AgMIP) compared the crop yield results for 14 global gridded crop models over
this century for maize, wheat, rice, and soybean (Elliott et al., 2015; Müller et al., 2017). The AgMIP study generally found





the negative effects of climate change (temperature and precipitation) on crop productivity (Müller et al., 2015), and enhanced photosynthesis and reduced water requirement by agroecosystems under increasing [$CO_2$] trends (Deryng et al., 2016). Studies also show that agriculture management practices generally can alleviate the impacts of climate change. For example, Levis et al. (2018) find irrigation is able to mitigate the yield losses due to increased respiration and water demand caused by temperature increase. Similarly, AgMIP models simulate higher yields under the assumption of no-N-limitation (Rosenzweig et al., 2014). However, the AgMIP study shows that the different models give results that are quite different under historical and future environmental change conditions (Rosenzweig et al., 2014).

Studies show interactions between the effects of environmental and management factors on crop yield. For instance, crop growth enhanced by the $CO_2$ fertilization effect is offset by low N supply and water stress conditions during vegetative growth (Ainsworth, 2008; Jain et al., 2009). Simultaneously, as the climate becomes warmer and damper, the inorganic soil N availability increases due to enhanced N mineralization associated with increased microbial decomposition and respiration rates (Jain et al., 2009; Rustad et al., 2001). Improved N availability and uptake lead to enhanced crop productivity. Thus the responses of interactions between C, water, temperature, and N input (fertilizer, manure and N deposition) would produce a stronger effect than the sum of their individual effects on crop yields. Therefore, crop models need to account for the synergistic effects of environmental factors and agricultural practices and how these effects control the yield of individual crops over the long-term.

The objective of this study is to address two specific questions: (1) what are the synergistic effects of environmental ([$CO_2$] and climate) and management (irrigation, nitrogen input, and harvest area) factors on historical maize (a C4 crop) and soybean (a C3 crop) yields, and (2) how do these effects change over the 21st century under two future scenarios for environmental and management factors? We address these two questions using a process-based land surface model, Integrated Science Assessment Model (ISAM). The model is driven with historical climate forcing data (1901-2015) and projected climate forcing data (2016-2100) for two Representative Concentration Pathways (RCP) 4.5 and 8.5. The other three model input variables – irrigation, N input, crop harvested areas – are developed here for two crops using the measurement data for historical time and two Shared Socio-Economic Pathways – SSP2 and SSP5 – to represent the agricultural activities under RCP4.5 and RCP8.5 scenarios, respectively (O'Neill et al., 2014; van Vuuren et al., 2011). We also evaluate our model results at regional (Figure S1) and global scale by comparison to measured data and other published model studies for the historical and future time periods.

## 2 Methods and Input Data

### 2.1 Model Description

ISAM, improved upon and used in this study, is a coupled biogeochemical and biogeophysical model with $0.5° \times 0.5°$ spatial resolution and multiple temporal resolutions ranging from half-hour to yearly time steps that simulates C, N, energy, and water budgets for various terrestrial ecosystems through the processes of photosynthesis, surface hydrology, radiative transfer,





carbon allocation, and ecosystem respiration (Barman et al., 2014a, 2014b; Yang et al., 2009). In addition, ISAM incorporates crop growth processes for C3 and C4 food crops (maize, soybean, wheat, and rice), and bioenergy grasses (miscanthus, cave-

in-rock, and alamo), which are evaluated at site-level, regional, and global scales (Lin et al., 2017; Niyogi et al., 2015; Song et al., 2013, 2015, 2016). Some of the important features, unique to ISAM and critical for crop yield and productivity calculations, include: (i) dynamic crop-specific phenology and carbon allocation schemes (Song et al., 2013, 2015), accounting for the sensitivity of different crops to extreme environmental conditions; (ii) dynamic vegetation structure, which better captures seasonal variability in LAI, canopy height, and root depth; (iii) dynamic root distribution processes at depth, to better

simulate root-mediated soil water uptake and transpiration. In the current study, we considered two crops, maize and soybean. In this study we extended ISAM to include (1) crop-specific planting time (Text S5), (2) crop-specific seeding rates (Text S6), (3) the curvature to the light response curve for the $CO_2$ fertilization effect (Text S7), (4) nutrient (e.g., N) stress while allocating the assimilated carbon to leaf, root, stem, and grain pools (Text S8), and extreme heat stress effect during crop productivity stage of the penology (Text S9)

ISAM has been extensively calibrated, validated, and evaluated for agricultural applications (Niyogi et al., 2015; Song et al., 2013, 2015, 2016) and in other different studies (Barman et al., 2014a, 2014b, 2016; El-Masri et al., 2013, 2015; Gahlot et al., 2017; Jain et al., 2006, 2013). In this study the modeled yields for two crops are further evaluated for elevated [$CO_2$], climate, N input and irrigation effects using free-air concentration enrichment (FACE) experiments and other site-specific data sets, and published model results at specific sites, regional and global scales over historical time and under future scenarios.

**2.2 Input Data**

*Environment Forcing Data*: Atmospheric $CO_2$ concentrations, and climate conditions for historical (1901-2015), and future (2016-2100) time periods are inputs for ISAM simulations of crop productivity. For the historical time period, we use yearly [$CO_2$] data from Meinshausen et al. (2011) and climate forcing data from CRU-NCEP (Harris et al., 2014; Viovy, 2016), which are available at a 6-hour time scale. The future calculations for crop productivity are performed for two climate scenarios: RCP

4.5-SSP2 and RCP8.5-SSP5 (O'Neill et al. 2014; van Vuuren et al., 2011). RCP4.5-SSP2 (hereafter referred to as RCP4.5) is a scenario to stabilize the total radiative forcing to 4.5 W/m² by the year 2100. In RCP4.5, economic, societal and technological trends are assumed similar to historical patterns.

RCP8.5-SSP5 (hereafter referred as RCP8.5) is a high energy demand scenario with relatively rapid economic development, causing high emissions and high greenhouse gas concentration scenario – representative of the highest GHG emissions

scenarios available in the literature – leading to total radiative forcing to 8.5 W/m² by the year 2100. ISAM simulations for the two scenarios use [$CO_2$] data from Meinshausen et al. (2011), and climate forcing data from a single ensemble member of NCAR's CESM model results (Levis et al., 2018; Ren et al., 2018) contributed to the CMIP5 effort (Meehl et al., 2012). The CESM model results are bias-corrected using the CRU-NCEP climate data as described in Text S1.



*Crop Specific Harvested Area*: The time-varying crop-specific annual harvested area at 0.5º x 0.5º was generated from combination of three data set, including global monthly irrigated and rainfed crop areas for the year 2000 (MIRCA2000, Portmann et al., 2010), global crop-specific harvested areas circa year 2000 (M3, Monfreda et al., 2008) and the Land-Use Harmonization 2 datasets (LUH2, Hurtt et al., in preparation). The process for calculating crop-specific harvested areas at a 0.5 x 0.5 spatial scale consists of multiple steps, as described in Text S2. In the RCP4.5 scenario, the maize and soybean harvested areas increase over the period 2016-2100; however, in the RCP8.5 scenario, areas for both crops increase until around 2040 and, thereafter, areas become stagnant until the end of 21 century (Figure S2a). The increase in future harvested lands is greater in regions that are currently developing, including Africa (AF), South and Southeast Asia (SSEA) and South America (SA). In contrast, the crop-specific harvest areas for maize and soybean decrease by the end of this century in NA, EU and the northeastern plain of CHN (Figure S3).

*Crop Specific N Input Amount*: Crop-specific annual spatial distribution of N input (fertilizer, manure, and atmospheric deposition) rates (kgN/ha) are calculated (Text S3), for historical time and the two future scenarios. To calculate the total N input at spatial scale we use the LUH2 datasets (Hurtt et al., in preparation) for N fertilizer data for historical and two future scenarios; Lamarque et al. (2011) and Tian et al. (2018) data for airborne nitrogen deposition (wet + dry) and M3 N input data for maize and soybean (Mueller et al., 2012) for the year 2000 in combination with the method described in Text S3. In the year 2000, the global N input amounts for maize and soybean are about 18 and 4 Tg N. The increasing trend of global N input coincides with the expansion of harvested areas over the past sixty years. For the future scenarios, the global average N application rates are higher and more prevalent in stronger N input areas under RCP4.5 compared to those under RCP8.5 conditions (Figure S2b). In addition, the magnitude of N usage is much higher for maize than for soybean. Higher input rates appear in higher production regions, including CHN, NA, and EU (Figure S4).

*Irrigation*: The cropland area of each grid cell is divided into irrigated and unirrigated according to the data for the irrigated fraction area for each grid cell (Text S2). On irrigated land, ISAM provides water when the root-zone soil water is limiting for crop photosynthesis, but the crop leaf area index is greater than zero as described in Supplementary Text S4. ISAM-estimated irrigation demand due to growing maize and soybean on irrigated croplands are approximately 47.6 km$^3$/yr and 8.0 km$^3$/yr, which fall within 11 global gridded crop models estimated range values (AgMIP, Müller et al., 2019) (Figure S5).

**2.3 Experimental Design and Analysis**

ISAM is spun-up by repeating the selected hourly climate forcing data (Harris et al., 2014; Viovy, 2016) for the period 1901-1920, and fixed year (1901) data for atmospheric $CO_2$ concentration of 296.8 ppm (Meinshausen et al., 2011), crop harvested areas and N input (see section 2.2) until the soil temperature and moisture and the carbon and nitrogen pools reach a steady state. The spin-up process is described in detail in El-Masri et al. (2015) and Song et al., (2016). Spin-up is followed by transient model simulations from 1901 to 2100 to calculates the productivity and yields for maize and soybean, as well as



corresponding C, N, water and energy fluxes by forcing the model with spatial-temporal varying climate forcing, N input data, crop harvested area sets and $[CO_2]$.

Six model simulations (Table 1) are carried out for the time period 1901-2100. The historical time-period simulation (1901-2005) is performed by forcing the model with historical data for different environmental and management factors, and the future time-period simulations (2006-2100) are forced with two future scenarios. These simulations are run to examine the effects of each individual factor over historical and future time periods (Table 1). In the reference case ($E_{Ref}$) all five factors vary with time over the time period 1901-2100. In three additional simulations, $E_{CO2}$, $E_{Cli}$, and $E_{Har}$, one of the five factors

remains fixed at the 1901 level for the historical time simulation and at the mean values for the time period, 1996-2005 level for the future scenario runs. In the other two simulations, $E_{Nit}$ and $E_{Irr}$, the N input and irrigation input are assumed zero. We then estimate the effect of each individual factor by differencing the yields between reference case and one of the four simulations: $CO_2$ fertilization ($E_{Ref} – E_{CO2}$), climate ($E_{Ref} – E_{Cli}$ ), irrigation ($E_{Ref} – E_{Irr}$), N input ($E_{Ref} – E_{Nit}$), and crop harvest area ($E_{Ref} – E_{Har}$). For the historical time period, we calculate the contribution of individual factor (in %) over a given time

(e.g., averaged over 1996-2005) relative to the $E_{Ref}$ case. In the case of the two future scenarios, we compare the results for the 2090s (e.g., averaged over 2090-2099) relative to 1996-2005. The results are masked out using the irrigated and rainfed harvest areas for each simulation at 0.5° x 0.5° (latitude x longitudes) spatial resolution. The total yield for each grid-cell is calculated by combining the weighted irrigated and rained yield together (Text S2). The results are presented at a spatial scale at 0.5° x 0.5° and a regional scale. To obtain the model results at regional scale we average the spatial results for each crop over its

cropland in six regions shown: North America (NA), South America (SA), Europe (EU), Africa (AF), China (CHN), and South and Southeast Asia (SSEA) (Figure S1).

## 3 Results

### 3.1 Model Estimated Crop Yields

#### 3.1.1 Yields for the historical time period

ISAM results for maize and soybean yields for the period 1996-2005 are compared to global and regional data sets available in literature in Figure 1, and on a 0.5$^0$ x 0.5$^0$ grid in Figure 2. ISAM results for higher maize yield regions (NA, western parts of EU, and northeastern CHN) and lower maize yield regions (India, Africa, and some areas in SA) are within the range of observed contemporary yields. ISAM also reproduces the measured pattern of soybean yields across high yield regions (NA, SA, and EU), and low yield regions (e.g., SSEA). The agreement between ISAM results and literature data is improved by the

implementation of additional processes and modifications in some existing processes in ISAM (Song et al., 2013), which are described next.





*Crop Specific Planting Time and Seeding Rates*. ISAM estimates of crop-specific planting time at a grid-scale (see Text S5), which are evaluated by comparison to literature data compiled by AgMIP (Elliott et al., 2015) in Figure S6. ISAM results show

that estimated planting time is controlled by climate conditions (Table S1) and different management practices (e.g., irrigation). In addition, we updated the crop seeding rates and residue amount (Text S6), which vary with season and planting conditions (Table S2). After implementing these modifications, modeled yield for soybean in CHN, AF, and SSEA regions are reduced and the revised yield in these regions for 1996-2005 average compare better with the observation data (Figures 1 and 2).

*CO2 Fertilization Effect.* The comparison of ISAM (Song et al., 2013) estimated yields with FACE sites (Table S3, Text S7 for FACE site calculations) results for crop productivity under elevated [$CO_2$] suggest that while the modeled yield for maize were consistent with FACE site results (Table S4), they were overestimated for soybean (Figure S7a, Table S4), because the electron transport rate calculations were not accounting the curvature to the light response curve, resulting in the overestimation of canopy temperature and the stomatal conductance (Figures S7b and S7c). However, after the implementation of curvature

to the light response curve (see detail description of the method and results in Text S7), ISAM estimated results under the irrigated or wet conditions are consistent with the measured values (Figure S7, Table S4). While the range of AgMIP model results under irrigated conditions is also comparable to the FACE experiment results, only ISAM is able to calculate the reduction in maize yield as observed in the FACE experiment (Table S4).

*N Stress Effect on Carbon Allocation.* Original ISAM model (Song et al., 2013) also overestimated the maize yields at the lower N application rates (Figure 8a, because the model overestimates the carbon allocated to the grain formation under the N stress conditions (i.e., the ratio of N supply and N demand). However, model results for soybean are consistent with the measured data (Figure S8b). After accounting the N stress effect on the carbon allocation to grain during initial and post-reproductive (grain-filling) stages of phenology (Text S8), Revise ISAM results for a lower N fertilizer rate for the six sites

(Table S5) show a stronger N stress, lower carbon allocation to grain formation compared to the results estimated based on the Original ISAM (Figure S8a). These results, which are consistent with the field experiment studies, suggest that maize growth slows down at the lower N supply rates, causing a decline in yield (Alemayehu et al., 2015; Gehl et al., 2005; Getachew and Belete, 2013; Hammad et al., 2011).

*Heat Stress (HS) Effect*. The heat stress shortens both the vegetative and reproductive phases of the crop phenology (Asseng et al., 2004; Teixeira et al., 2013) resulting in a decrease of the average growing season length (GSL) and crop yields (See Section 4.2). We implemented HS in ISAM by accounts the impact of heat stress on reduction in carbon allocation during the reproductive of the vegetative and reproductive phases of the phenology. While other studies consider air temperature for the calculation of HS effect (Deryng et al., 2014; Teixeira et al., 2013), the crop canopy temperature is shown to better explain

yield reductions associated with heat stress (e.g., Gabaldón-Leal et al., 2016; Siebert et al., 2014; Webber et al., 2017), which we consider in our model simulations (Text S9).



Model results show that biases in modeled yields across all regions (Table 2) are reduced after implementations of various modifications as discussed above. Overall, the percent bias (PBIAS, Text S10) results show the model estimated yields at global and regional scales are compared well with the observations after model improvements, with the exception of maize

(16%) in SSEA and soybean (-36%) in CHN and SA (27%) (Table 2). The PBIAS for maize yield is reduced from -29% to 3% on a global scale. The model results for soybean yield are also improved, particularly for the tropical regions CHN, AF, and SSEA (Table 2). The remaining model biases might be due to the model limitations in estimating nutrients limitation; crop mortality effects due to ozone, wind, hail, weeds, pests, and disease, and/or due to not accounting for the cropping systems in the model.

In addition, ISAM is able to reproduce the observed (FAOstat, 2017) detrended global and regional yields (Text S11 describe the method to calculate detrended yield) over the period 1982-2006 with the correlation coefficient, $r$, 0.66 for maize and 0.56 for soybean (Figure S9). These values are close to the middle of the range of values estimated based on the ensemble of 14 global AgMIP model results; 0.42-0.89 for maize and 0.37-0.67 for soybean (Müller et al., 2017). Model estimated detrended yields for both crops at the regional scale are also compared well with FAOstat (2017), with the exception of the values of $r$

for soybean in AF, SSEA, and CHN and for maize in SSEA where ISAM estimated values are lower than FAOstat (2017). The previous modeling studies (e.g., Müller et al., 2017) have also used the FAOstat data for their model evaluations and noticed the same inconsistency and suggested that this might be related to the reporting year issue; some crop yields harvested at the end of the calendar year are reported by FAOstat (2017) in the following year report. Therefore, the detrended FAOstat yield might have a one-year delay in contrast to the values of ISAM in some years.

**3.1.2 Crop yield under RCP4.5 and RCP8.5 scenarios**

ISAM-estimated changes in global crop yield, are driven by environmental and management factors specified by scenarios RCP4.5 and RCP8.5. Changes in yield in the 2090$_S$ relative to that in 1996-2005 are shown in Table S6 and Figure 3. For scenario RCP8.5, estimated maize yield is projected to decrease across all regions except for EU and AF. For scenario RCP4.5, maize yield is projected to increase in all regions, except for CHN. In contrast, soybean yield increases across all regions under

both scenarios, except for the EU under RCP4.5 (Figure 3 and Figure S10).

**3.2 The Effects of Changes in Environmental and Management Factors and on Maize and Soybean Yields**

Each of the four environmental and management factors considered ($CO_2$, climate, N input, and irrigation) result in an estimated increase in yield of maize and soybean at the global-scale from 1901 to 1996-2005 (see Table S6 and Figure S11a). The yields for both crops increase across all regions due to the $CO_2$ fertilization effect, but the increase is stronger for soybean

than for maize, because the photosynthesis for soybean is relatively less saturated under ambient [$CO_2$] (McGrath and Lobell, 2013). Without the [$CO_2$] increase, the global maize and soybean yields are lower by 4% and 16%, respectively (Table S6). Over the last century climate has a small positive global effect (2 and 4%), with some regions showing a positive and others a negative effect (see Table S6 and Figure S11a).



Out of the five factors studied here (including a harvested area for future scenarios), N input increases the historical and future
maize yields under two scenarios the most, [$CO_2$] affects the soybean yield under the two future scenarios (see Table S6 and
Figure 4). On the other hand, climate and crop harvested area changes decrease the yields for both crops under the two scenarios.
On a regional scale, N input continues to be a strong contributor to yields in the 2090s for maize for all regions under both
scenarios, whereas irrigation shows slightly positive effect across all regions and harvested area mixed effect, in some regions
positive and in others negative, for both crops for the two scenarios (see Figure 4).

## 4 Discussion

### 4.1 Model Evaluation Using Data

Global and regional crop yields estimated with the ISAM land surface model for the C3 crop soybean and the C4 crop maize
are consistent with literature data averaged over the period 1996-2005 (Figures 1 and 2). ISAM-calculated yield variability at
regional and global scales over the longer time period, 1982-2006, are also consistent with FAOstat (2017) (Figure S9). In
addition to this study, the overall confidence in ISAM estimated yields for maize and soybean is strengthened by the validation
of ISAM results at the site level (Song et al., 2013) and the country level (Niyogi et al., 2015).

### 4.2 Estimated Effects of Environmental and Management Factors

ISAM results show that over the past century and for two future scenarios, RCP 4.5 and RCP 8.5, environmental and
management factors affect maize and soybean yields:
*$CO_2$ Fertilization*. We find that the modeled $CO_2$ fertilization is stronger for soybean across all regions than for maize, because
ISAM's net photosynthesis rate increases due to higher carboxylation rates and lower of photorespiration rates for soybean
than for maize. The effect is stronger in tropical regions (SA, AF, and SSEA) (Table S6 and Figure 11a), because of (1) greater
availability of nitrogen via N fixing bacteria (seen in measurements in Ainsworth et al. 2002), (2) smaller leaf area index (LAI)
(Figure S12), which absorbs higher photosynthetically active radiation (PAR), because PAR is inversely proportional to LAI
(seen in measurements in Sakurai et al. 2014), and (3) higher temperatures enhance the $CO_2$ fertilization effect on net
photosynthesis rate, because with rising temperature both the maximum carboxylation rate of Rubisco (seen in measurements
in Bernacchi et al. 2006; Ruiz-Vera et al. 2013). In contrast, the $CO_2$ fertilization effect for maize is higher in semi-arid and
temperate rainfed agro-climatic regions (e.g., NA, EU, AF, and CHN) for the period 1996-2005 (Table S6, Figure S11a), where
dryer soil conditions lead to partially closed stomata thus reducing transpiration and soil water stress on maize but also decrease
access to $CO_2$ (seen in measurements in Leakey et al. 2006). Similar to the historical case, the increase in yield due to the $CO_2$
fertilization effect at regional and global scales in the 2090s is stronger for soybean than for maize under both future scenarios
(Figures 4, S11b and S11c).
We compared ISAM-estimated changes of global-average yield in 2080 for both crops for with (w/) and without (w/o) $CO_2$
cases under RCP8.5 with model results available in the literature – AgMIP (Deryng et al., 2016) and NCAR's CLM model



(Ren et al., 2018) – in Table 3. While ISAM-estimated maize and soybean yields w/CO$_2$ falls within the interquartile range of AgMIP model results, there are large differences in the spatial patterns (not shown here), mainly because ISAM includes the effects of extreme conditions on crop phenology, carbon allocation, and structures growth (e.g. leaf area, canopy height, and root depth) (Song et al., 2013), such as the abiotic effects of cold and warm temperatures, frost damage and heat stress, during initial and post-reproductive processes. ISAM also accounts for N limitation on grain filling during the reproductive phase of
phenology. These effects are rarely considered in other modeling studies.

*Climate Change (Excluding Heat Stress).* The climate effect on yield over the past century differs by region (Table S6 and Figure S11), but in line with previous studies (e.g., Challinor et al., 2014; Fodor et al., 2017; Rosenzweig et al., 2014). The hotter temperatures over most of the tropical regions reduce yields for both crops, even though these regions experience higher
precipitation. In contrast, increased temperatures enhanced yields for both crops in colder regions (NA, eastern EU, northeastern CHN, and in boreal latitude zones) (Figure S11a), where moderate warming increased the length of the growing period (LGP). For the future scenarios, the effect of climate lower yields of both crops in all regions, with a stronger effect on soybean than maize (Figure 4), because the optimum leaf temperature for photosynthesis is higher for maize than for soybean. For example, net photosynthesis rate, *A,* for maize (soybean) increases up to 50 µmol CO$_2$/m$^2$/s (25 µmol CO$_2$/m$^2$/s) with the
leaf temperature increases to 40°C (25°C); thereafter *A* decreases with increasing leaf temperatures (Figure S13).

*Heat Stress.* ISAM estimated results show a higher global heat stress effect during the vegetative and reproductive stages on maize (-2.7% and -5.0% under RCP4.5 and RCP8.5) than on soybean (-2.4 % and -4.9% under RCP4.5 and RCP8.5) by the 2090s (Figure S14). While these results are consistent with other modeling results (e.g., Deryng et al., 2014; Teixeira et al.,
2013), ISAM simulations are performed using canopy temperature, which are ameliorated by the cooling effect of irrigation and decreases the yield losses due to heat stress. The published modeling studies, on the other hand, use much warmer prescribed air temperature (as opposed to canopy temperature) for the heat stress calculations. ISAM results indicate that South Asia, Sahel, Eastern China, Spain, parts of Central Asia (e.g., Russian Federation), Central NA, and Eastern Brazil, and Central SA are regions of high heat stress for maize (Figure S14). A high heat stress effect on soybean yield is found in Central NA
and SA, Northern India, Eastern CHN, and Southwest region of Russia (Figure S14). ISAM-estimated global patterns of heat stress on crops are consistent with published studies on the heat stress-related global agricultural hot-spots regions (Gourdji et al., 2013; Teixeira et al., 2013).

*Irrigation.* Irrigation enhances maize yield more than that of soybean over the last century and under the two future scenarios
(Table S6), because of a higher fraction of irrigated area for maize than for soybean. The global maize yield with irrigation is estimated to increase by about 7%, whereas only 2% for soybean over 1996-2005 (Table S6). These results are consistent with previous studies (Chen et al., 2018; Irwin et al., 2017; Kranz & Benham, 2001; Verma et al., 2005). On a regional scale, the effects of irrigation over 1996-2005 is most obvious in temperate and semi-arid areas for both crops, including central and



western parts of NA, northeastern CHN, Eastern Australia, Middle East, Central Asia, and western EU (Figure S11a). However,
soybean yield under irrigated case does not change much in the tropical agriculture regions, e.g., SA and SSEA (Table S6),
because of low irrigated harvested areas and higher precipitation rates. However, the global and region yields change due to
irrigation do not change much by the 2090s under the future scenarios, with the exception of maize yield in SSEA under
RCP4.5 and soybean in EU under RCP8.5 (Table S6 and Figure 3).

*N Input*. The effect of N input is stronger for maize than for soybean because soybean is an N-fixing crop (Table S6 and Figure
S11). The stimulation of yield with N input is greater in N-limiting and high N-application regions, including NA, EU, and
CHN. These regions are water-limiting regions too. Therefore, irrigation amplifies the N input effect by reducing the water
stress effect and enhancing the root carbon that stimulates more N uptake (Yang et al., 2009). The interactive effect of N under
both scenarios increases the yield for both crops by 2090s in most of the regions, especially under the RCP4.5 scenario with a
higher N input amount and improved N availability under favorable environmental conditions (Figure 4). The responses are
lower under RCP8.5 not only because of lower nitrogen input rates per harvest area, but also due to warmer conditions,
weakening the N input effect (Figure S2b).

*Harvest Area Change.* The effect of variation of crop harvested area changes on crop yield under future scenarios is estimated
using time-varying crop harvested areas (Figures S2a and S3). The yield difference for $E_{Ref} - E_{Har}$ case at the global and
regional scales can be explained by the relative changes in the crop production and harvested areas over the 21$^{st}$ century (Figure
S15). The estimated yield is lower when the relative change in the harvest area is greater than the relative change in crop
production. For example, global maize harvested area increased by 38% by 2090s under RCP4.5, but production increase is
only 16% (Figure S15) under RCP4.5 by the 2090s, which causes about a 20% reduction in yield (Table S6, Figure 4). The
yield for both crops is reduced for most of the regions by the 2090s under two scenarios for the same reasons, except in SA
for maize under RCP4.5 and for both crops under RCP8.5, in EU and AF for soybean under RCP8.5, and in SSEA for both
crops under both scenarios, because production is increased more or reduced less than harvested areas (Figure 15).

*Co-limitation Effects*. ISAM results confirm that management factors help to offset some of the negative effects of climate
change and limitations of resources (e.g., water and fertilizer). For example, over the historical time irrigation and N input
offset a decrease in crop yields otherwise caused by drier and N-limitation conditions, and increases in yield from $CO_2$
fertilization (Table S6). However, ISAM results also show that crop productivity is co-limited by environmental factors, such
as [$CO_2$] and climate. Therefore, management factors under the higher [$CO_2$] and warmer future climate scenario RCP8.5 may
not be able to offset all of the crop yields losses by the end of this century. This is the case, for example, for global maize yield
w/management case under the RCP8.5 scenario where yield is estimated to be about 20% lower in the 2090s than in 1996-
2005 (Figure 3).

*Summary and Future Directions*. This study's results show the importance of five environmental and management factors on model-estimated productivity of maize and soybean. Looking beyond this study, new studies should improve the basic

understanding of the interconnected processes of crop water use, C, N, and other nutrients and implement the processes of ozone damage, weeds, and pests in crop growth models, in order to provide more accurate future projections of crop yield. In addition, more experiment-based studies are needed to investigate crop yield responses to extreme events, such as drought, flood, wind damage, and heatwaves, to improve the representation of various stress constraints in crop models. New studies should also improve the treatment of management practices so that observed yield trends can be simulated and future trend

projections can be understood (c.f. Alexandratos and Bruinsma, 2012). For example, the irrigation algorithm in ISAM does not consider different irrigation sources (such as irrigated water from groundwater pumping), methods and pathways (flood irrigation, drip irrigation, and sprinkler irrigation) which affect irrigation water use efficiency and hydrological cycles (Leng et al., 2017). Thus, improving water use processes for each crop type in the model has the potential to improve the estimation of global human water usage, crop yield and irrigation demand for crops (Webber et al., 2016). Furthermore, approaches to

include the effects of changing technology and management practices (e.g. crop cultivars and pest/weed management) on both the simulation of past yield change and the projection of future yield remains a challenge for the land process models like ISAM.

**Data availability**. We provide the following data sets:

(1) Spatial distribution of mean maize and soybean yield (unit: ton/ha for 0.5° x 0.5° grid) from ISAM simulations and literature averaged for the time period 1996-2005 (Figure 2). (2) Spatial distribution of maize and soybean harvested areas (unit: hectare) for 1901-2100. The data for the period 2016-2100 are based on RCP 4.5 and RCP 8.5 scenarios. (3) Spatial distribution of N fertilizer and manure amount (unit:kgN/ha) applied over the maize and soybean harvested areas for 1901-2100. The data for 2016-2100 are based on RCP 4.5 and RCP 8.5 scenarios. (4) ISAM simulated spatial distribution of maize and soybean yields

(unit: t/ha) based on various model experiments, which are described in Table 1.

All the data are stored in NetCDF files. The data can be accessed from the ISAM website: http://climate.atmos.uiuc.edu/Lin_Cropyields/

**Supplement**. The supplement related to this article is available online at: https://doi.org/XXX.


**Author contributions**. AKJ and TSL conceived the study and drafted the manuscript. PL and HSK provided intellectual input and extensively edit the manuscript. TSL and YS collected and analyzed model input data, and performed ISAM model simulations.

**Competing interests**. The authors declare that they have no conflict of interest.



**Acknowledgments**. This work is supported by the U.S. National Science Foundation (NSF-AGS-12-43071). We would also like to acknowledge high-performance computing support from Cheyenne (doi:10.5065/D6RX99HX) provided by NCAR's Computational and Information Systems Laboratory, sponsored by the National Science Foundation.




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





**Figures and Tables**

**Table 1.** Model experiment design to study the effects of individual environmental and management factors over the period 1901-2100. Tick mark (✓) indicates the factor was varied with time. Cross mark (✗) indicates the factor was held fixed over time. For the historical and future simulations with fixed N Input or Irrigation, factor values remain fixed at 1901 values (zero N inputs and irrigation assumed). For the climate fixed case, the climate data for the period 1901-1920 was repeated over the

period 1901-2005 and recycle the climate data of 1996-2005 from 2006 onward. In the case of fixed harvest area, the crop harvest area remained fixed at the mean of 1996-2005 for the time-period 2006-2100. For the fixed $[CO_2]$ case, the $[CO_2]$ remain fixed at 1901 level (296.8 ppm) for the period 1901-2015 and at mean of 1996-2005 (368.2 ppm) for the period 2006-2100.

| Cases | $[CO_2]$ | Climate | N Input | Irrigation | Harvest Area |
|-------|----------|---------|---------|------------|--------------|
| **Experiments to Study Sensitivity to Environmental and Management Factors** | | | | | |
| $E_{Ref}$ | ✓ | ✓ | ✓ | ✓ | ✓ |
| $E_{CO2}$ | ✗ | ✓ | ✓ | ✓ | ✓ |
| $E_{Cli}$ | ✓ | ✗ | ✓ | ✓ | ✓ |
| $E_{Nit}$ | ✓ | ✓ | ✗ | ✓ | ✓ |
| $E_{Irr}$ | ✓ | ✓ | ✓ | ✗ | ✓ |
| $E_{Har}$ | ✓ | ✓ | ✓ | ✓ | ✗ |





**Table 2.** Global and regional-scale percent bias (PBIAS, %) of maize and soybean yields from ISAM (Original and Revised versions) and the average of various measured data for the period 1996-2005[1].

| Global/Region | Maize[2] | | Soybean[3] | |
|---|---|---|---|---|
| | Song et al. (2013) | This Study | Song et al. (2013) | This Study |
| Global | -29.2 | 3.1 | -28.7 | 5.6 |
| North America (NA) | -18.9 | 8.3 | -24.8 | -3.1 |
| South America (SA) | -64.0 | -5.1 | -12.0 | 27.1 |
| Europe (EU) | -26.8 | 3.2 | -6.0 | 14.5 |
| Africa (AF) | -80.3 | -5.3 | -94.0 | 5.3 |
| China (CHN) | -15.4 | -7.8 | -79.4 | -35.8 |
| South and South East Asia (SSEA) | -22.1 | 16.0 | -96.1 | 9.7 |

[1]The measured data set are the average of various data, including Iizumi et al. (2014) for the period 1996-2005, Monfreda et al. (2008, M3) for the year 2000 , You et al. (2014, MapSPAM2000 & 2005) for year 2000 and 2005, and FAOstat (2017) for period 1996-2005.

[2]The Original and Revised columns are the % bias (PBIAS) for w/o and w/ N stress effect on carbon allocation for maize (Supplementary Text S8).

[3]The Original and Revised columns are the % bias (PBIAS) for w/o and w/ N stress effect on carbon allocation (Supplementary Text S8), seeding rates (Supplementary Text S6), and elevated $CO_2$ effect (Supplementary Text S7) for soybean.






**Table 3.** Comparison of ISAM estimated global maize and soybean yields change to that of other model results available in the literature: percent change in average yield from 1996-2005 (or 2000) to 2076-2085 (or 2080) under the RCP8.5 scenario. The results are compared for reference w/ $CO_2$ ($E_{Ref}$, Table 1) and w/o $CO_2$ ($E_{CO2}$, Table 1) cases. The ISAM w/o $CO_2$ case is the same as $E_{CO2}$, except that $CO_2$ remains fixed at 2005 level of 380 ppm. The nitrogen input is applied as per the $E_{Ref}$ and $E_{CO2}$ experiments. The ISAM yield results are weighted by fixed irrigated and rainfed harvested areas.

| Crop | ISAM | | CLM* | | AgMIP** | |
|---|---|---|---|---|---|---|
| | w/$CO_2$ | w/o $CO_2$ | w/$CO_2$ | w/o $CO_2$ | w/$CO_2$ | w/o $CO_2$ |
| Maize | -0.2 | -8.1 | 0.7 | –9.2 | [–16.4; 1.0] | [–28.2; –13.3] |
| Soybean | 28.7 | -12.8 | 18.6 | –9.6 | [–12.1; 33.3] | [–40.5; –27.7] |

*Results are taken from Ren et al. (2018). N fertilization application is set at North American crop-specific levels everywhere and fixed over time. Then yield is adjusted with nitrogen fertilizer assumptions based on FAO data.

**Results are the interquartile range across all six global gridded crop models run with climate date from five global climate models (Deryng et al., 2016). The EPIC, GEPIC and pDSSAT models apply fertilizer dynamically through the crop growing season: application occurs at specific stages of the crop development to take into account the role of both application quantity and timing. PEGASUS applies fertilizer as a daily stress function and thus does not simulate the effect of fertilizer directly. LPJmL and LPJ-GUESS do not represent fertilizer application.

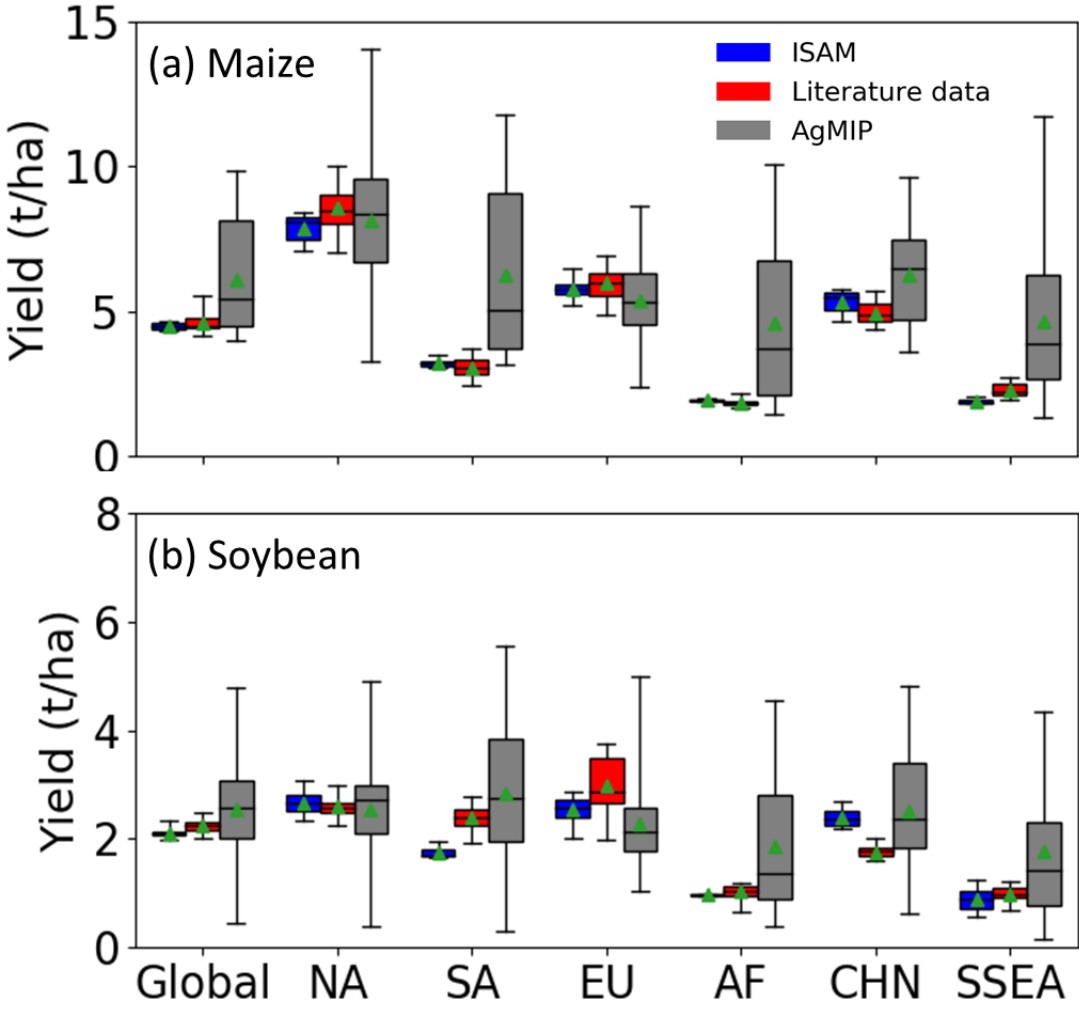

**Figure 1.** Global and regional-scale comparsions of (a) maize and (b) soybean annual yield (t /ha of global's/region's harvested area) from ISAM, AgMIP (Müller et al., 2019), and the available data set values in the literature. Model results are for the period 1996-2005 and the literature data include: Iizumi et al. (2014) for years 1996-2005, Monfreda et al. (2008, M3) for 2000, You et al. (2014) for year 2000 and 2005, and FAOstat (FAOstat, 2017) for years 1996-2005. The AgMIP results are for 12 different crop models. The boxes are the interquartile ranges, the horizontal lines plotted in the boxes are the median values, and the whiskers indicate the highest and lowest values of the results. The green triangles marked in the boxes are the mean values.



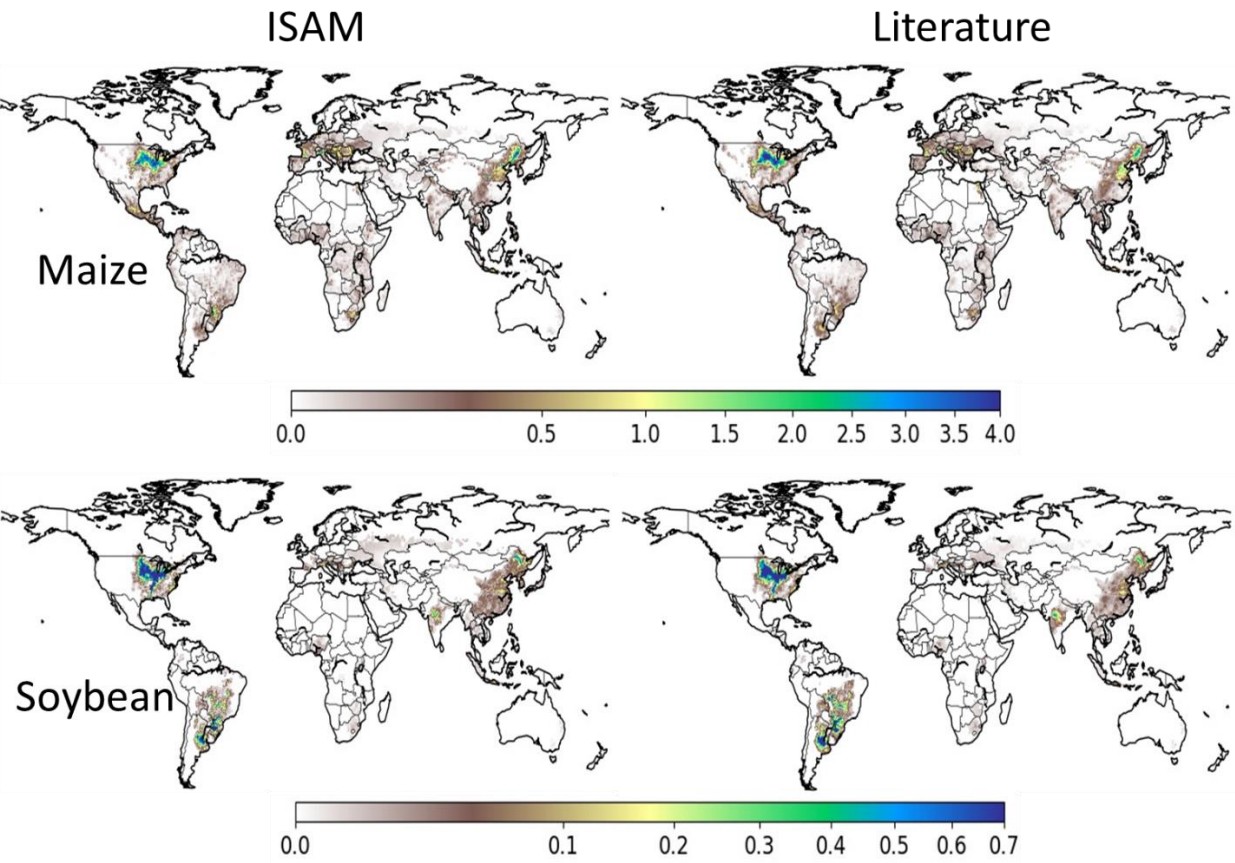

595 **Figure 2.** Comparison of ISAM estimated spatial distributions of maize and soybean yields (ton /ha of 0.5° x 0.5° grid-cell)
with referenced data during year 1996-2005. The maize and soybean from ISAM model is weighted by irrigated and rainfed
harvested areas and averaged from year 1996 to 2005. The harvested areas for both crops are masked by crop-specific harvested
area. Literature data set are the average of Iizumi et al. (2014) for the period 1996-2005, Monfreda et al. (2008) for year 2000,
and You et al. (2014) for the year 2000 and 2005.

600



**Figure 3.** (a) Maize and (b) soybean yield changes (%) at regional and global scales for the 2090s average relative to the 1996-2005 average under RCP 4.5 (green bars) and RCP 8.5 (brown bars) scenarios. Solid bars are results for with (w/) varying environmental and management factors based on future scenarios and crosshatched bars are results w/ varying environmental factors, but without (w/o) varying management factors, assuming harvest area remains unchanged from the average value for 1996–2005, and nitrogen input and irrigation are curtailed after 2000.









**Figure 4.** Maize and soybean yield contributions (%) for the average over the 2090s (2090-2099) at global and regional scales due to the effects of $CO_2$, climate, nitrogen input, irrigation, and crop harvested area under RCP4.5 and RCP8.5 scenarios ($E_{Ref}$ in 2090s minus $E_{XXX}$ in 2090s then divided by $E_{Ref}$ in 1996-2005). $E_{XXX}$ are the factor experiments shown in Table 1.