# Peer review of "Effects of environmental and management factors on worldwide maize and soybean yields over the 20[th] and 21[st] centuries"

_Biogeosciences, 2020_

## Short Comment (SC1) · 26 Jun 2020

**Corrected Equation, Table, and Figures appearing in the Supplementary Section**

**Updated equation (SE8a) appearing in Text S5**

$$\sigma_y = \sqrt{\frac{1}{12-1} \times \sum_{m=1}^{12} \left( \overline{Z_{m,y}} - \mu_y \right)^2}$$   (SE8a)

> **Commented [JA1]:** This has to be squared

**Updated Table S4**

Table S4. ISAM estimated change (%) in maize and soybean yields due to elevated [$CO_2$] concentration are compared with FACE site data and CLM and AgMIP model results under wet/irrigated conditions[1].

| Crop | FACE | Model | | |
|---|---|---|---|---|
| | | ISAM[4] | CLM[7] | AgMIP[7] |
| Maize | -1.0[2] | -0.03 | 0.9 | 0.2 - 7.7 |
| Soybean | 14.4 ± 12.5[3] | 17.5± 2.7[5] (32.1± 4.3)[6] | 22 | 5.2 - 44.4 |

> **Commented [JA2]:** This is the correct value

[1] The positive values are net sink of C by the terrestrial ecosystem
[2] German- FACE site
[3] SoyFACE site
[4] See supplementary Text S7 for ISAM simulations
[5] Calculated by the revised version of ISAM
[6] Calculated by the original version of ISAM
[7] Deryng et al. (2016) (See Text S11 for CLM and AgMIP models simulations)

[Figure]

**Updated Figure S11b and S11c with the correct results for Nitrogen and Irrigation effects on Maize and Soybean Yields**

Figure S11b. Same as Figure S12a, but the changes (%) from 1996-2005 and to 2090s under RCP4.5.

[Figure]

Figure S11c. Same as Figure S11a, but the changes (%) from 1996-2005 and to 2090s under RCP8.5.

---

## Referee Comment (RC1) · Anonymous Referee #1 · 19 Aug 2020

Lin et al. Effects of enviornmnetal and management factors on world-wide maize and soybean yields over the 20th and 21st centuries.

This manuscript reported past-present-future crop production with focus on maize and soybean using a numerical model simulation. The authors used ISAM model (which is widely applied in many studies) to simulate production with time-variable inputs of climate, atmospheric CO2, N input, irrigation, and harvest area at global scale from 1901 to 2100. Changes in climate and human activities and its effects on crop production is important topic, and model projection work is clearly important.

There're some existing and collaborative studies, called AgMIP (and others), as listed in

this study. This study tries to go further from AgMIP, and try to understand interactions between environmental (natural) factors and human factors. Target of the study is good, and relevant to BG.

However, when I read the manuscript, there're some concerns and a substantial amount of major comments. At the present stage, I cannot recommend it for publication. Even if my comments to model setup is incorrect, this manuscript requires re-read, re-write, and re-check by co-authors. It requires substantially large modification from beginning to end.

Major comments

Paper Preparation.

The manuscript is not ready for submission. There are many typos and simple mistakes. I can easily find (e.g. grammatically incorrect) such sentences. I strongly suggest the authors check this manuscript from beginning to end and improve description. In addition, too many figures (15 figures and many tables) in supplemental materials are cited in the result section (Much of results section are supported by supplemental figures). Therefore, it is unfair as a peer-reviewed articles. These supplemental figures are referred from the main text but main text lacks sufficient explanation and interpretation of the figure. These should be improved.

Model Setup

In the experiments of sensitivity check, the authors conducted many experiment with one input time-invariant. In the experiment, the authors set 1901-1920 values for 1901-2005 simulation and 1996-2006 values for 2006-2100 simulation. If the description is true, I have serious concern of the model outputs. If we consider atmospheric $CO_2$ concentration as time-invariant parameter, model run was conducted using 1901-1920 $CO_2$ concentration for 1901-2005 and 1996-2006 $CO_2$ for 2006-2100. This setup include large jump of $CO_2$ concentration, and this may introduce unrealistic jumps of

outputs around 2005. So, in my guess, results on future changes contain large biases caused by jump of input parameters for sensitivity tests. This should be avoided.

Interpretation of Results

Manuscript contains many figures (in main text and supplement). However, these figures were not well-evaluated and discussed in the main text. I believe these figures contains many important implications. However, this manuscript fails on this point.
* * *

---

## Referee Comment (RC2) · David Lapola (Referee) · 20 Aug 2020

Although the impact of climate change on future crop yields has been subject to a large number of previous studies (e.g. Lobell et al. 2008 Science), the presented manuscript is innovative in terms that it considers the different influences of not only climate but also management practices. It is not an incredible discover but deserves publication given the well-organized and concise presentation.

I have only a few minor comments:

Abstract —> it should be finished with the implications of these results to climate

change adaptation in the agricultural sector.

L118: Please provide the reference for the irrigated land extension in addition to the reference to Text S2.

L220: Isn't the fact that maize is a C4 grass and soybean is a C3 plant a more precise explanation for this?

L225: I wonder if the yield increase due to CO2 fertilization wouldn't change nutritional contents (e.g. C:N ratios) of harvested parts of soybean.(?)

L246: The sentence ends abruptly, please revise.

L256: Statements like "not shown here" are increasingly less recommended by scientific journals like Biogeosciences. I suggest this spatial comparison is shown in the supplementary material.

————————————————

---

## Author Comment (AC1) · 7 Sep 2020

Reviewers' comments are in **bold**, the authors' responses are in normal font, and the suggested changes for the text in *italics*.

We thank the reviewer for recognizing the importance of the work presented in this study, in particular our crop modeling efforts to understand interactions between environmental factors (natural) and management factors (human).

**However, when I read the manuscript, there're some concerns and a substantial amount of major comments. At the present stage, I cannot recommend it for publication. Even if my comments to model setup is incorrect, this manuscript requires re-read, re-write, and re-check by co-authors. It requires substantially large modification from beginning to end.**

We completely understand the reviewer's concern. We will thoroughly go throw the MS and improve the readability further by better streamlining the paper.

We provide our detailed responses to the major comments below.

**Major comments**

**Paper Preparation.**

**The manuscript is not ready for submission. There are many typos and simple mistakes. I can easily find (e.g. grammatically incorrect) such sentences. I strongly suggest the authors check this manuscript from beginning to end and improve description.**

We understand the reviewer's concern regarding typos and simple mistakes. We apologize for these oversights. We propose to proofread the manuscript again and improve its readability.

**In addition, too many figures (15 figures and many tables) in supplemental materials are cited in the result section (Much of results section are supported by supplemental figures). Therefore, it is unfair as a peer-reviewed articles. These supplemental figures are referred from the main text but main text lacks sufficient explanation and interpretation of the figure. These should be improved.**

Originally, we moved those figures, tables, as well as some technical information to the supplementary section (SS), because they were providing extra details related to the individual aspects of the study but not critical to support the conclusion of the study findings, and whose inclusion in the main text would have disrupted the flow of

the descriptions of the results. However, we appreciate the reviewer's concerns that the SS contains too many figures and other material, which would be published as a separate document along with the paper. To address the reviewer's concern, here we propose the following changes: (1) move most of the technical information and some figures and tables to appendices, (2) retain some of the material in the SS, and (3) move some to the main text. See below our detailed plan:

*(1) Material to be Moved to Appendix Section* According to the Biogeosciences policy, appendices are part of the manuscript, which are published right after the conclusions and discussion sections. In specific, we propose to move the following supplementary text sections, figures, and tables to the Appendix Section.

(1.1) Merge e following four text sections (Text S2-S5) into one Appendix B:

Text S2. Estimation of Crop Specific Harvested Area for Irrigated and Non-Irrigated Conditions Text S3. Estimation of Crop Specific N Inputs at Spatial Scale Text S4. Estimation of Irrigation Water Amount Text S5. Estimation of Crop Specific Planting Time

Appendix B: Estimation of Crop Specific Harvested Area, N Input, Irrigation Amount, and Planting Time We will also move Figure S5 and Table S1, which are cited in Text S4 and Text S5, to Appendix B

(1.2) Move the following text sections to the Appendix Section to standalone Appendices:

Text S1. to an Appendix Section with the following title: Appendix A: Bias Correction of Future Climate

Text S6 to an Appendix Section with the following title: Appendix C. Seeding and Plant Residue Removal Rates We will also move Table S2, which is cited in Text S6, to Appendix C

Text S7 to an Appendix Section with the following title: Appendix D. ISAM Model Simulations Yields for Maize and Soybean for FACE Sites We will also move Figure S7 and Table S3, which are cited in Text S7, to Appendix D

Text S8 to an Appendix Section with the following title: Appendix E. Implementation of the N Stress Effect on Carbon Allocation We will also move Figure S8 and Table S5, which are cited in Text S8, to Appendix E

Text S9 to an Appendix Section with the following title: Appendix F. Heat Stress effect on Crop Productivity

Text S10 to an Appendix Section with the following title: Appendix G. The Calculation of the Percent Bias (PBIAS)

Text S11 to an Appendix Section with the following title: Appendix H. Calculation of Detrended Yield We will also move Figure S9, which is cited in Text S11, to Appendix H

Text S12 to an Appendix Section with the following title: Appendix I. CLM and AgMIP Model Results for the FACE Sites We will also move Tables S4, which is cited in Text S12, to Appendix I

Appendix J: Figures Figure S12: It shows the model estimated LAI for the period 1996-2005 and for the 2090s under two future scenarios. While the observations suggest that LAI for soybean for the tropical regions is the lowest, we added this figure to show that the model estimated LAI for soybean is consistent with the observations. We propose to move this figure to the Appendix section.

Figure S13: This figure shows the model simulated leaf net photosynthetic rates for C4 crops, maize and C3 crops, soybean response to leaf temperature, which are important results. Therefore, we propose to move this figure to the Appendix section

*(2) Material to be Retained in the Supplementary Section* We will retain the following Text, Tables, and Figures in the supplementary section because these are not critical to supporting the conclusion of the paper.
2.1 Figures Figures S1-S4, S6: These figures describe the model input data, which we adopted from other published studies.

Figures S10-S11a-c: These figures are the expansion of Figures 3 and 4. Figures 3-4 describe the results at a regional scale, which are further expended to the gridded scales in Figures S10-S11ac.

Table S6: The table shows the maize and soybean yields (t/ha) at global and regional scales averaged over the period 1996-2005 for the reference case (ERef) and for the [CO2] (ECO2), climate (ECli), irrigation (EIrr), nitrogen input (ENit) and harvest areas (EHar) factor cases; and the
*(3) Material to be Moved to the Main Text* Figures 14S and 15S: These figures are showing the model estimated effect of heat stress and harvested area change on crop yield under two scenarios. These figures are cited multiple times in the main text. Therefore, we propose to move these figures to the main text.

**Model Setup In the experiments of sensitivity check, the authors conducted many experiment with one input time-invariant. In the experiment, the authors set 1901-1920 values for 1901- 2005 simulation and 1996-2006 values for 2006-2100 simulation. If the description is true, I have serious concern of the model outputs. If we consider atmospheric CO2 concentration as time-invariant parameter, model run was conducted using 1901-1920 CO2 concentration for 1901-2005 and 1996-2006 CO2 for 2006-2100. This setup include large jump of CO2 concentration, and this may introduce unrealistic jumps of outputs s around 2005. So, in my guess, results on future changes contain large biases caused by jump of input parameters for sensitivity tests. This should be avoided.**

We would like to clarify the procedure of model simulations. We propose to revise the text in the paper to describe the model simulation procedure better.

First, we agree that increasing the concentrations to 1996-2005 value in the year 2005 in the historical invariant CO2 case would have introduced an unrealistic jump in the

model output in the year 2001(2005). However, this is not our modeling approach. We perform two different CO2 invariant cases, one is a historical case, and another one is a future case as explained below:

For the historical CO2 invariant case, we run the model from 1901 to 2005 with fixed CO2 concentration at the 1900 level.

For the future invariant CO2 case, we first run the model for the reference case for the period 1901-2000; the reference case has a varying climate, CO2 and other environmental and management inputs. Then we continue the model run from 2001-2100 with CO2 concentration fixed at the 1996-2005 mean value (ca. 369.0 ppm), which is ca. 2000 value (368.2), but other variable values are assumed to change as in the reference case. Note that we compare the results for the 2090s (e.g., averaged over 2090-2099) relative to 1996-2005 (ca. 2000), which has already been stated in the main text. Following this approach, there is no sudden jump in the model output for the yields in the year 2000 (or 2005), and the year 2000 model results are the same for CO2 invariant and reference cases.

We will revise the text from lines 135-137 and clarify this point in the revised MS as follows:

*The five additional simulations, ECO2, ECli, EHar, ENit and EIrr are performed differently for the period 1901-2005, and for the period 2006-2100. For the 1901-2005case, one of the five factors remains fixed at the 1901 level, whereas all other factors vary with time as in ERef. For the future time case, first we run the model for the ERef case for the period 1901-2000. Next, we continue the model run from 2001-2100 with one of the five factors remains fixed at the 1996-2005 mean value, which represents ca. 2000 value, but other variable values are assumed to change as in the ERef case.*

**Interpretation of Results   Manuscript contains many figures (in main text and supplement). However, these figures were not well-evaluated and discussed in the main text. I believe these figures contains many important implications. How-**

**ever, this manuscript fails on this point.**

We agree that there are lots of important results that can be drawn from these figures. However, we have discussed only those aspects of these figures, which are directly or indirectly related to the objectives of this study. However, we have re-evaluated each figure and table discussion, and propose to add the following additional explanations as we find appropriate.

Figure S4: This figure shows the future changes in the N input at the regional scale. We propose to add the following text in line 112 to describe the N input distribution at a regional scale based on this figure:

*For future scenarios, the global average N application rates are higher and more prevalent in greater increased harvested areas, including SA, AF, and SSEA, under RCP4.5 compared to those under RCP8.5 conditions (Figure S2b and S4). In the 2090s, N rate is decreased in NA and EU under RCP8.5; however, it is increased in EU and decreased in NA under RCP4.5. In CHN, there is a negligible change under RCP4.5 but an increase in the south under RCP8.5.*

Table S2: These tables describe the changes in seeding rates and residue management. Since their revised input, as shown in Table S2, improve modeled crop yields at regional scales (Table 2), particularly in AF and SSEA), we propose to add the following text in lines 162 describe these improvements:

*The updated seeding rates at the sowing time are usually lower for soybean in CHN, AF and SSEA (Table S2). After implementing these modifications, the modeled yield for soybean is reduced and the revised yield in these regions for 1996-2005 compares better with the observation data (and the biases in the modeled yield relative to the historical time are reduced (Figures 1 and 2).*

Table 2: This table shows the global and regional-scale percent bias (PBIAS,

*Also, uncertainty in the input data, such as climate, soil, or crop management, might*

have also introduced the biases in the modeled yield (Barman et al., 2014a, 2014b; Kheshgi et al., 1999; Jagtap and Jones, 2001), which we plan to carry out in our future modeling analysis.

Figure 4: This figure describes how different environmental and management factors, including climate change, will affect crop yields at a regional scale. Here we will add the following statement in line 270 to describe how the climate affects maize and soybean yield (Figure 4) differently through crop respiration:

*Also, rising temperature increases crop respiration and thus reduces carbon use efficiency (CUE), defined as the ratio of net primary production to gross primary production (Zhang et al., 2013). Since CUE is lower for soybean than maize (Yamaguchi, 1978), soybean incurs relatively higher carbon loss through respiration, resulting in the lower yields under the higher emission scenarios RCP 8.5 (Figure 4).*

Figures 1 and 4: Figure 1 shows the current (1996-2005) maize and soybean yields in each region. Figure 4 shows how N input affecting yield over the 2090s under two scenarios in each region. Here we propose to add the following text in line 302 to describe further the effects of N management on future crop yields for different regions:

*Crop yield can be enhanced by the intensification of N fertilization with the expansion of harvested areas in current low crop productive regions, including AF and SSEA for maize and soybean, and SA for maize (Figure 1). In AF there is a continuous loss of soil fertility and N mining, agriculture practices resulting in higher N losses than N added to the soils (Vitousek et al. 2009, Liu et al. 2010; Lassaletta et al. 2014).*

---

## Author Comment (AC2) · 7 Sep 2020

Reviewers' comments are in **bold**, the authors' responses are in normal font, and the suggested changes for the text in *italics*.

**Although the impact of climate change on future crop yields has been subject to a large number of previous studies (e.g. Lobell et al. 2008 Science), the presented manuscript is innovative in terms that it considers the different influences of not only climate but also management practices. It is not an**

**incredible discover but deserves publication given the well-organized and concise presentation.**

The authors would like to thank the Anonymous Reviewer 2 for his/her valuable comments and suggestions to strengthen the analysis presented in our manuscript. We also thank the reviewer for the encouraging comments about our study.

**I have only a few minor comments: Abstract: it should be finished with the implications of these results to climate change adaptation in the agricultural sector.**

Thank you for your suggestion. Here we propose to add the following statement in the abstract.

*Moreover, climate change adaptations through N management and irrigation will benefit crop yield, particularly for the maize, under the higher emission scenario.*

**L118: Please provide the reference for the irrigated land extension in addition to the reference to Text S2.**

We propose to add references in the following line:

*The irrigated fraction area for each grid cell is assigned based on three datasets (Hurtt et al. 2020; Monfreda et al. 2008; Portmann et al. 2010) (Text S2).*

**L220: Isn't the fact that maize is a C4 grass and soybean is a C3 plant a more**

**precise explanation for this?**

We thank the reviewer for this point and revise the text as follows:

*The yields for both crops increase across all regions due to the CO2 fertilization effect, but the increase is stronger for soybean than for maize, because soybean is a C3 crop and maize C4 crop. Therefore, photosynthesis for soybean is relatively less saturated under ambient [CO2] (McGrath and Lobell, 2013).*

**L225: I wonder if the yield increase due to CO2 fertilization wouldn't change nutritional contents (e.g. C:N ratios) of harvested parts of soybean.(?)**

It is suggested that there is not much change in the C:N ratios of the harvested part of the soybean due to CO2 fertilization under growth chamber conditions (Zheng et al. 2020).

To address your points, we will add the following statement:

*It has been found that the C:N ratios of the harvested part of the soybean do not change much due to CO2 fertilization under growth chamber conditions (Zheng et al. 2020).*

**L246: The sentence ends abruptly, please revise.**

We revise the sentence as follows:

*higher temperatures enhance the CO2 fertilization effect on net photosynthesis rate, because with rising temperature, both the specificity of Rubisco for CO2 and solubility*

*of CO2 in water decline relative to O2 (Bernacchi et al. 2006; Ruiz-Vera et al. 2013).*

**L256: Statements like "not shown here" are increasingly less recommended by scientific journals like Biogeosciences. I suggest this spatial comparison is shown in the supplementary material.**

Thank you for your suggestion. We will add the following figure to the Appendix section.

Figure SX is attached as a Supplement Document

Figure SX. The spatial pattern of ISAM (a-b) and AgMIP maize and soybean yield changes (percent) under the RCP8.5 scenario in 2076-2085 (or ca. 2080) relative to 1996-2005. AgMIP estimated are the median (c-d) values for the 30 ensemble runs (six crop models results based on five climate model forcing data) (Deryng et al., 2016). The crop areas are masked based on the MIRCA2000 data set.

REFERENCES CITED

Bernacchi, C. J., Leakey, A. D. B., Heady, L. E., Morgan, P. B., Dohleman, F. G., Mc-Grath, J. M., Gillespie, K. M., Wittig, V. E., Rogers, A., Long, S. P. and Ort, D. R.: Hourly and seasonal variation in photosynthesis and stomatal conductance of soybean grown at future CO2 and ozone concentrations for 3 years under fully open-air field conditions, Plant, Cell Environ., 29(11), 2077–2090, doi:10.1111/j.1365-3040.2006.01581.x, 2006.

Hurtt, G. C., Chini, L., Sahajpal, R., Frolking, S., Bodirsky, B. L., Calvin, K., Doelman, J. C., Fisk, J., Fujimori, S., Goldewijk, K. K., Hasegawa, T., Havlik, P., Heinimann, A., Humpenöder, F., Jungclaus, J., Kaplan, J., Kennedy, J., Kristzin, T., Lawrence, D., Lawrence, P., Ma, L., Mertz, O., Pongratz, J., Popp, A., Poulter, B., Riahi, K., Shevliakova, E., Stehfest, E., Thornton, P., Tubiello, F. N., van Vuuren, D. P., and Zhang, X.: Harmonization of Global Land-Use Change and Management for the Period 850–2100 (LUH2) for CMIP6, Geosci. Model Dev. Discuss., https://doi.org/10.5194/gmd-2019-360, in review, 2020.

McGrath, J. M. and Lobell, D. B.: Regional disparities in the $CO_2$ fertilization effect and implications for crop yields, Environ. Res. Lett., doi:10.1088/1748-9326/8/1/014054, 2013. Monfreda, C., Ramankutty, N. and Foley, J. A.: Farming the planet: 2. Geographic distribution of crop areas, yields, physiological types, and net primary production in the year 2000, Global Biogeochem. Cycles, 22(1), 1–19, doi:10.1029/2007GB002947, 2008. Portmann, F. T., Siebert, S. and Döll, P.: MIRCA2000-Global monthly irrigated and rainfed crop areas around the year 2000: A new high-resolution data set for agricultural and hydrological modeling, Global Biogeochem. Cycles, doi:10.1029/2008gb003435, 2010.

Ruiz-Vera, U. M., Siebers, M., Gray, S. B., Drag, D. W., Rosenthal, D. M., Kimball, B. A., Ort, D. R. and Bernacchi, C. J.: Global warming can negate the expected $CO_2$ stimulation in photosynthesis and productivity for soybean grown in the midwestern United States, Plant Physiol., 162(1), 410–423, doi:10.1104/pp.112.211938, 2013.

Zheng, G., Chen, J. and Li, W.: Impacts of $CO_2$ elevation on the physiology and seed quality of soybean, Plant Diversity, 42(1), 44-51, doi:10.1016/j.pld.2019.09.004, 2020.

**Supplement:**

---

## Author Comment (AC3) · 7 Sep 2020

AUTHORS' RESPONSE TO COMMENTS BY THE REVIEWERS

Reviewers' comments are in **bold**, the authors' responses are in normal font, and the suggested changes for the text in *italics*.

**Lin et al. Effects of enviornmnetal and management factors on world-wide maize and soybean yields over the 20th and 21st centuries. This manuscript reported**

**past-present-future crop production with focus on maize and soybean using a numerical model simulation. The authors used ISAM model (which is widely applied in many studies) to simulate production with time-variable inputs of climate, atmospheric CO2, N input, irrigation, and harvest area at global scale from 1901 to 2100. Changes in climate and human activities and its effects on crop production is important topic, and model projection work is clearly important. There're some existing and collaborative studies, called AgMIP (and others), as listed in this study. This study tries to go further from AgMIP, and try to understand interactions between environmental (natural) factors and human factors. Target of the study is good, and relevant to BG.**

The authors would like to thank the Anonymous Reviewer 1 for his/her valuable comments and suggestions to strengthen the analysis presented in our manuscript. We also thank the reviewer for recognizing the importance of the work presented in this study, in particular our crop modeling efforts to understand interactions between environmental factors (natural) and management factors (human).

**However, when I read the manuscript, there're some concerns and a substantial amount of major comments. At the present stage, I cannot recommend it for publication. Even if my comments to model setup is incorrect, this manuscript requires re-read, re-write, and re-check by co-authors. It requires substantially large modification from beginning to end.**

We completely understand the reviewer's concern. We will thoroughly go throw the MS and improve the readability further by better streamlining the paper.

We provide our detailed responses to the major comments below.

**Major comments**
**Paper Preparation.**

**The manuscript is not ready for submission. There are many typos and simple mistakes. I can easily find (e.g. grammatically incorrect) such sentences. I strongly suggest the authors check this manuscript from beginning to end and improve description.**

We understand the reviewer's concern regarding typos and simple mistakes. We apologize for these oversights. We propose to proofread the manuscript again and improve its readability.

**In addition, too many figures (15 figures and many tables) in supplemental materials are cited in the result section (Much of results section are supported by supplemental figures). Therefore, it is unfair as a peer-reviewed articles. These supplemental figures are referred from the main text but main text lacks sufficient explanation and interpretation of the figure. These should be improved.**

Originally, we moved those figures, tables, as well as some technical information to the supplementary section (SS), because they were providing extra details related to the individual aspects of the study but not critical to support the conclusion of the study findings, and whose inclusion in the main text would have disrupted the flow of the descriptions of the results. However, we appreciate the reviewer's concerns that the SS contains too many figures and other material, which would be published as a separate document along with the paper. To address the reviewer's concern, here we propose the following changes: (1) move most of the technical information and some figures and tables to appendices, (2) retain some of the material in the SS, and (3) move some to the main text. See below our detailed plan:

*(1) Material to be Moved to Appendix Section* According to the Biogeosciences policy, appendices are part of the manuscript, which are published right after the conclusions and discussion sections. In specific, we propose to move the following supplementary text sections, figures, and tables to the Appendix Section.

(1.1) Merge e following four text sections (Text S2-S5) into one Appendix B:

Text S2. Estimation of Crop Specific Harvested Area for Irrigated and Non-Irrigated Conditions Text S3. Estimation of Crop Specific N Inputs at Spatial Scale Text S4. Estimation of Irrigation Water Amount Text S5. Estimation of Crop Specific Planting Time

Appendix B: Estimation of Crop Specific Harvested Area, N Input, Irrigation Amount, and Planting Time We will also move Figure S5 and Table S1, which are cited in Text S4 and Text S5, to Appendix B

(1.2) Move the following text sections to the Appendix Section to standalone Appendices:

Text S1. to an Appendix Section with the following title: Appendix A: Bias Correction of Future Climate

Text S6 to an Appendix Section with the following title: Appendix C. Seeding and Plant Residue Removal Rates We will also move Table S2, which is cited in Text S6, to Appendix C

Text S7 to an Appendix Section with the following title: Appendix D. ISAM Model
Simulations Yields for Maize and Soybean for FACE Sites We will also move Figure S7 and Table S3, which are cited in Text S7, to Appendix D

Text S8 to an Appendix Section with the following title: Appendix E. Implementation of the N Stress Effect on Carbon Allocation We will also move Figure S8 and Table S5, which are cited in Text S8, to Appendix E

Text S9 to an Appendix Section with the following title: Appendix F. Heat Stress effect on Crop Productivity

Text S10 to an Appendix Section with the following title: Appendix G. The Calculation of the Percent Bias (PBIAS)

Text S11 to an Appendix Section with the following title: Appendix H. Calculation of Detrended Yield We will also move Figure S9, which is cited in Text S11, to Appendix H

Text S12 to an Appendix Section with the following title: Appendix I. CLM and AgMIP Model Results for the FACE Sites We will also move Tables S4, which is cited in Text S12, to Appendix I

Appendix J: Figures Figure S12: It shows the model estimated LAI for the period 1996-2005 and for the 2090s under two future scenarios. While the observations suggest that LAI for soybean for the tropical regions is the lowest, we added this figure to show that the model estimated LAI for soybean is consistent with the observations. We propose to move this figure to the Appendix section.

Figure S13: This figure shows the model simulated leaf net photosynthetic rates for

C4 crops, maize and C3 crops, soybean response to leaf temperature, which are important results. Therefore, we propose to move this figure to the Appendix section

*(2) Material to be Retained in the Supplementary Section* We will retain the following Text, Tables, and Figures in the supplementary section because these are not critical to supporting the conclusion of the paper.

2.1 Figures Figures S1-S4, S6: These figures describe the model input data, which we adopted from other published studies.

Figures S10-S11a-c: These figures are the expansion of Figures 3 and 4. Figures 3-4 describe the results at a regional scale, which are further expended to the gridded scales in Figures S10-S11ac.

Table S6: The table shows the maize and soybean yields (t/ha) at global and regional scales averaged over the period 1996-2005 for the reference case (ERef) and for the [CO2] (ECO2), climate (ECli), irrigation (EIrr), nitrogen input (ENit) and harvest areas (EHar) factor cases; and the

*(3) Material to be Moved to the Main Text* Figures 14S and 15S: These figures are showing the model estimated effect of heat stress and harvested area change on crop yield under two scenarios. These figures are cited multiple times in the main text. Therefore, we propose to move these figures to the main text.

**Model Setup In the experiments of sensitivity check, the authors conducted many experiment with one input time-invariant. In the experiment, the authors set 1901-1920 values for 1901- 2005 simulation and 1996-2006 values for 2006-2100 simulation. If the description is true, I have serious concern of the model outputs. If we consider atmospheric $CO_2$ concentration as time-invariant parameter, model run was conducted using 1901-1920 $CO_2$ concentration for 1901-2005 and 1996-2006 $CO_2$ for 2006-2100. This setup include large jump of $CO_2$ concentration, and this may introduce unrealistic jumps of outputs s around 2005. So, in my guess, results on future changes contain large biases caused by jump of input parameters for sensitivity tests. This should be avoided.**

We would like to clarify the procedure of model simulations. We propose to revise the text in the paper to describe the model simulation procedure better.

First, we agree that increasing the concentrations to 1996-2005 value in the year 2005 in the historical invariant $CO_2$ case would have introduced an unrealistic jump in the model output in the year 2001(2005). However, this is not our modeling approach. We perform two different $CO_2$ invariant cases, one is a historical case, and another one is a future case as explained below:

For the historical $CO_2$ invariant case, we run the model from 1901 to 2005 with fixed $CO_2$ concentration at the 1900 level.

For the future invariant $CO_2$ case, we first run the model for the reference case for the period 1901-2000; the reference case has a varying climate, $CO_2$ and other environmental and management inputs. Then we continue the model run from 2001-2100

with CO2 concentration fixed at the 1996-2005 mean value (ca. 369.0 ppm), which is ca. 2000 value (368.2), but other variable values are assumed to change as in the reference case. Note that we compare the results for the 2090s (e.g., averaged over 2090-2099) relative to 1996-2005 (ca. 2000), which has already been stated in the main text. Following this approach, there is no sudden jump in the model output for the yields in the year 2000 (or 2005), and the year 2000 model results are the same for CO2 invariant and reference cases.

We will revise the text from lines 135-137 and clarify this point in the revised MS as follows:

*The five additional simulations, ECO2, ECli, EHar, ENit and EIrr are performed differently for the period 1901-2005, and for the period 2006-2100. For the 1901-2005case, one of the five factors remains fixed at the 1901 level, whereas all other factors vary with time as in ERef. For the future time case, first we run the model for the ERef case for the period 1901-2000. Next, we continue the model run from 2001-2100 with one of the five factors remains fixed at the 1996-2005 mean value, which represents ca. 2000 value, but other variable values are assumed to change as in the ERef case.*

**Interpretation of Results  Manuscript contains many figures (in main text and supplement). However, these figures were not well-evaluated and discussed in the main text. I believe these figures contains many important implications. However, this manuscript fails on this point.**

We agree that there are lots of important results that can be drawn from these figures. However, we have discussed only those aspects of these figures, which are directly or

indirectly related to the objectives of this study. However, we have re-evaluated each figure and table discussion, and propose to add the following additional explanations as we find appropriate.

Figure S4: This figure shows the future changes in the N input at the regional scale. We propose to add the following text in line 112 to describe the N input distribution at a regional scale based on this figure:

*For future scenarios, the global average N application rates are higher and more prevalent in greater increased harvested areas, including SA, AF, and SSEA, under RCP4.5 compared to those under RCP8.5 conditions (Figure S2b and S4). In the 2090s, N rate is decreased in NA and EU under RCP8.5; however, it is increased in EU and decreased in NA under RCP4.5. In CHN, there is a negligible change under RCP4.5 but an increase in the south under RCP8.5.*

Table S2: These tables describe the changes in seeding rates and residue management. Since their revised input, as shown in Table S2, improve modeled crop yields at regional scales (Table 2), particularly in AF and SSEA), we propose to add the following text in lines 162 describe these improvements:

*The updated seeding rates at the sowing time are usually lower for soybean in CHN, AF and SSEA (Table S2). After implementing these modifications, the modeled yield for soybean is reduced and the revised yield in these regions for 1996-2005 compares better with the observation data (and the biases in the modeled yield relative to the historical time are reduced (Figures 1 and 2).*

Table 2: This table shows the global and regional-scale percent bias (PBIAS,

*Also, uncertainty in the input data, such as climate, soil, or crop management, might have also introduced the biases in the modeled yield (Barman et al., 2014a, 2014b; Kheshgi et al., 1999; Jagtap and Jones, 2001), which we plan to carry out in our future modeling analysis.*

Figure 4: This figure describes how different environmental and management factors, including climate change, will affect crop yields at a regional scale. Here we will add the following statement in line 270 to describe how the climate affects maize and soybean yield (Figure 4) differently through crop respiration:

*Also, rising temperature increases crop respiration and thus reduces carbon use efficiency (CUE), defined as the ratio of net primary production to gross primary production (Zhang et al., 2013). Since CUE is lower for soybean than maize (Yamaguchi, 1978), soybean incurs relatively higher carbon loss through respiration, resulting in the lower yields under the higher emission scenarios RCP 8.5 (Figure 4).*

Figures 1 and 4: Figure 1 shows the current (1996-2005) maize and soybean yields in each region. Figure 4 shows how N input affecting yield over the 2090s under two scenarios in each region. Here we propose to add the following text in line 302 to describe further the effects of N management on future crop yields for different regions:

*Crop yield can be enhanced by the intensification of N fertilization with the expansion of harvested areas in current low crop productive regions, including AF and SSEA for maize and soybean, and SA for maize (Figure 1). In AF there is a continuous loss*

*of soil fertility and N mining, agriculture practices resulting in higher N losses than N added to the soils (Vitousek et al. 2009, Liu et al. 2010; Lassaletta et al. 2014). We consider 85 percent removal of residue at the harvest time in these regions (Text S6) as N management practice, and our model result shows that yields for both crops are increased under RCP4.5, but not under the RCP8.5 scenario (Figure 4 and 11; Table S6).*

Figure S15: This figure shows future production and harvested areas changes relative to 2000 conditions at the regional scale. It has an important implication on future global maize and soybean production. We will add the text in lines 312 in the revised MS as follows:

*It is also important to note that total crop production (maize and soybean) in AF has the largest increase among all regions under both scenarios. These scenario results are consistent with a study by Foyer et al. (2018), which suggests that the crop production in AF to increase because of growing demand for the crops.*

REFERENCES CITED

Bernacchi, C. J., Leakey, A. D. B., Heady, L. E., Morgan, P. B., Dohleman, F. G., McGrath, J. M., Gillespie, K. M., Wittig, V. E., Rogers, A., Long, S. P. and Ort, D. R.: Hourly and seasonal variation in photosynthesis and stomatal conductance of soybean grown at future CO2 and ozone concentrations for 3 years under fully open-air field conditions, Plant, Cell Environ., 29(11), 2077–2090, doi:10.1111/j.1365-3040.2006.01581.x, 2006.

Hurtt, G. C., Chini, L., Sahajpal, R., Frolking, S., Bodirsky, B. L., Calvin, K., Doelman, J. C., Fisk, J., Fujimori, S., Goldewijk, K. K., Hasegawa, T., Havlik, P., Heinimann, A., Humpenöder, F., Jungclaus, J., Kaplan, J., Kennedy, J., Kristzin, T., Lawrence, D.,

Lawrence, P., Ma, L., Mertz, O., Pongratz, J., Popp, A., Poulter, B., Riahi, K., Shevli-akova, E., Stehfest, E., Thornton, P., Tubiello, F. N., van Vuuren, D. P., and Zhang, X.: Harmonization of Global Land-Use Change and Management for the Period 850–2100 (LUH2) for CMIP6, Geosci. Model Dev. Discuss., https://doi.org/10.5194/gmd-2019-360, in review, 2020.

McGrath, J. M. and Lobell, D. B.: Regional disparities in the CO2 fertilization effect and implications for crop yields, Environ. Res. Lett., doi:10.1088/1748-9326/8/1/014054, 2013.

Monfreda, C., Ramankutty, N. and Foley, J. A.: Farming the planet: 2. Geographic distribution of crop areas, yields, physiological types, and net primary production in the year 2000, Global Biogeochem. Cycles, 22(1), 1–19, doi:10.1029/2007GB002947, 2008.

Portmann, F. T., Siebert, S. and Döll, P.: MIRCA2000-Global monthly irrigated and rainfed crop areas around the year 2000: A new high-resolution data set for agricultural and hydrological modeling, Global Biogeochem. Cycles, doi:10.1029/2008gb003435, 2010.

Ruiz-Vera, U. M., Siebers, M., Gray, S. B., Drag, D. W., Rosenthal, D. M., Kimball, B. A., Ort, D. R. and Bernacchi, C. J.: Global warming can negate the expected CO2 stimulation in photosynthesis and productivity for soybean grown in the midwestern United States, Plant Physiol., 162(1), 410–423, doi:10.1104/pp.112.211938, 2013.

Zheng, G., Chen, J. and Li, W.: Impacts of CO2 elevation on the physiology and seed quality of soybean, Plant Diversity, 42(1), 44-51, doi:10.1016/j.pld.2019.09.004, 2020.

---

## Author Comment (AC4) · 7 Sep 2020

This is the revised Supplementary Section submitted by Tzu-Shun Lin, the first author of the paper. We don't have to add any additional comments to make on this document.